# Revisiting Few-Shot Object Detection with Vision-Language Models

**Anish Madan**[1,*] **Neehar Peri**[1,*], **Shu Kong**[2,3,†] **Deva Ramanan**[1,†]

[1]Carnegie Mellon University, [2]University of Macau, [3]Institute of Collaborative Innovation

## Abstract

The era of vision-language models (VLMs) trained on web-scale datasets challenges conventional formulations of "open-world" perception. In this work, we revisit the task of few-shot object detection (FSOD) in the context of recent foundational VLMs. First, we point out that zero-shot predictions from VLMs such as GroundingDINO significantly outperform state-of-the-art few-shot detectors (48 vs. 33 AP) on COCO. Despite their strong zero-shot performance, such foundation models may still be sub-optimal. For example, `trucks` on the web may be defined differently from `trucks` for a target application such as autonomous vehicle perception. We argue that the task of few-shot recognition can be reformulated as aligning foundation models to target concepts using a few examples. Interestingly, such examples can be multi-modal, using both text and visual cues, mimicking instructions that are often given to human annotators when defining a target concept of interest. Concretely, we propose Foundational FSOD, a new benchmark protocol that evaluates detectors pre-trained on any external data and fine-tuned on multi-modal (text and visual) K-shot examples per target class. We repurpose nuImages for Foundational FSOD, benchmark several popular open-source VLMs, and provide an empirical analysis of state-of-the-art methods. Lastly, we discuss our recent CVPR 2024 Foundational FSOD competition and share insights from the community. Notably, the winning team significantly outperforms our baseline by 23.3 mAP! Our code and dataset splits are available on GitHub and HuggingFace.

## 1 Introduction

Vision-language models (VLMs) trained on (often proprietary) web-scale datasets have disrupted traditional notions of the "open-world", particularly for few-shot recognition. In this paper, we revisit few-shot object detection (FSOD) in the context of these foundation models, propose a new benchmark protocol that allows foundation models to "enter the conversation", and present several simple baselines.

First, we highlight that *zero-shot* predictions from VLMs like GroundingDINO [33] demonstrate a remarkable improvement over state-of-the-art *few-shot* detectors (48.3 vs. 33.1 AP) on COCO [31], as shown in Table 1. In hindsight, this is not surprising, as the former is pre-trained on far more data (that may include visual examples of the target concept), while the later is pre-trained on data that is explicitly curated to avoid target concepts of interest. From this perspective, VLMs violate the current training protocol of few-shot benchmarks, suggesting that such protocols need to be rethought in the foundational era.

**Concept Alignment.** Despite their impressive performance, foundation models used in a zero-shot fashion can still be sub-optimal. For example, `trucks` as defined for a particular target application

---

[*]Equal Contribution
[†]Equal Senior Authorship

38th Conference on Neural Information Processing Systems (NeurIPS 2024) Track on Datasets and Benchmarks.

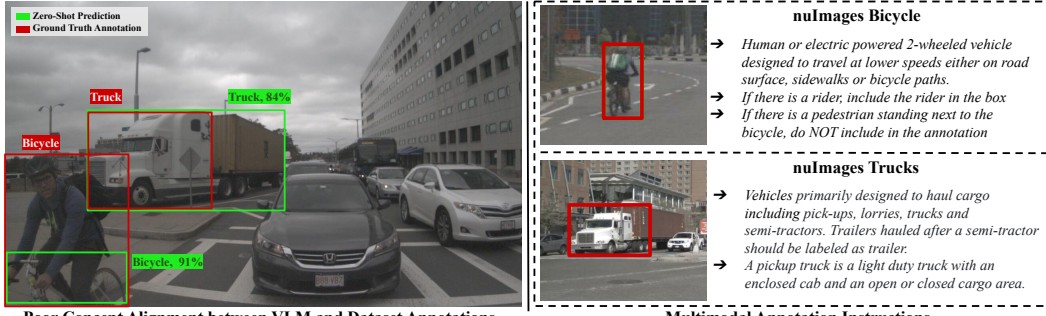

**Poor Concept Alignment between VLM and Dataset Annotations**       **Multimodal Annotation Instructions**

Figure 1: **Poor Alignment Between Vision Language Models (VLMs) and Target Concepts**. Although VLMs show impressive zero-shot performance, they struggle when the target class is different from concepts encountered in web-scale training. On the **left**, we see that the nuImages dataset [2] defines the cab of the `truck` as a separate concept from its `trailer` (shown in **red**). In contrast, the VLM predicts the entire vehicle as a `truck` (shown in **green**). Similarly, nuImages annotations dictate that a person riding a bicycle must also be labeled as part of `bicycle` (shown in **red**) unlike the VLM prediction (in **green**). On the **right**, we present the actual *class definitions* given to the nuImages annotators, provided as both textual descriptions and visual examples. Just as human annotators learn concepts from few-shot multi-modal examples, we argue that VLMs should be aligned with $K$ vision-language examples.

like perception for autonomous vehicles may differ from `trucks` as found on the web (cf. Fig. 1). Indeed, this well-known observation has created the ad-hoc practice of prompt engineering, where users actively search for a textual prompt that elicits the desired zero-shot behaviour. Instead, we argue that one can principally address the challenge of *aligning* foundation models to target concepts through the lens of few-shot recognition, by presenting VLMs with a few examples of the target concept. Crucially, such examples can be multi-modal, using both text and visual cues, mimicking the natural few-shot *multi-modal instructions* that are often given to human annotators when defining a target concept of interest [3]. Before introducing our new protocol, we first review the conventional FSOD setup below.

**Conventional FSOD.** Existing FSOD benchmarks partition object detection datasets like PASCAL VOC [8] and COCO [31] into `base` and `novel` classes. Detectors pre-train on `base` classes and then learn to identify `novel` classes given $K$ examples (or $K$-shots). Current protocols enforce `base` and `novel` to be disjoint to prevent concept leakage, allowing one to evaluate generalization to the "open-world". However, as most detectors are pre-trained on ImageNet [5], we point out that *concept leakage already occurs in the current FSOD protocol*. For example, `cat` and `person` are deemed `novel` for COCO-FSOD but are present in ImageNet data used to pre-train detectors [49]. Moreoever, `car` is deemed `novel`, but similar concepts like `sports car` and `race car` are present in ImageNet, illustrating the difficulty of even defining leakage.

**Foundational FSOD.** We believe that concept leakage should be embraced. Our Foundational FSOD protocol replaces the `base` pre-training stage with web-scale pre-training, where such data may be proprietary and not fully disclosed [44]. *We argue that pre-training on large-scale data will be the key enabler for generalization to the open world*. Note that this hypothesis is difficult to even evaluate under the conventional few-shot protocol, motivating our setup. Moreover, another key property is that $K$-shot instances may include multi-modal examples spanning both images and text, motivating a multi-modal adaptation stage that aligns the VLM to target concepts (cf. Fig. 2). We repurpose nuImages [2], a challenging dataset due to open-world categories such as `debris` and `pushable-pullable`, for our Foundational FSOD benchmark.

We present three major contributions.

- We modernize FSOD benchmarks by embracing foundational VLMs that are pretrained on internet-scale data. We highlight the practical challenge of using multi-modal few-shot examples to define target semantic concepts (as shown in Fig. 1).

- We adapt nuImages for Foundational FSOD, evaluate popular open-source VLMs, and present an empirical analysis of leading methods.

- We highlight the results from our recent CVPR 2024 challenge hosted in conjunction with the Workshop on Visual Perception via Learning in An Open World.

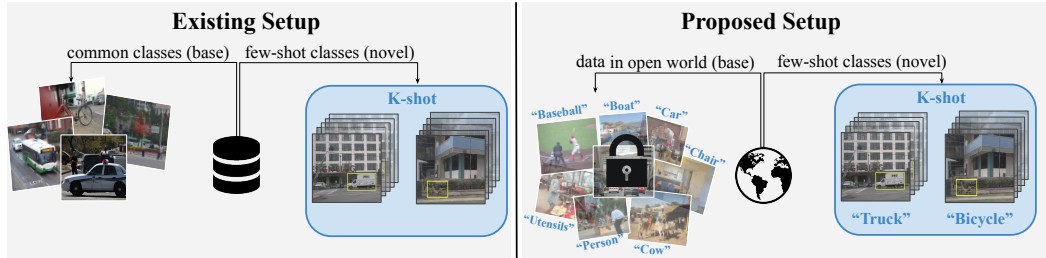

Figure 2: **Foundational Few-Shot Object Detection (FSOD)**. Conventional FSOD protocols (**left**) allow for pre-training on `base` classes (with many examples per class) and then fine-tuning on $K$-shots of `novel` classes, where `novel` and `base` are designed to be disjoint. However, we point out that pre-training datasets such as ImageNet often contain classes similar to `novel` classes, highlighting the issue of concept leakage. As preventing concept leakage is difficult (if not impossible) and appears to be artificial in the foundational era, we propose *Foundational FSOD* (**right**). Our setup allows for pre-training on massive (and potentially proprietary) datasets, typical for foundational vision-language models. Since these models can process both text and images, one can utilize such *multi-modal K*-shot examples to *align* VLMs with the target concepts of interest.

## 2 Related Works

**Few-Shot Object Detection** aims to detect new categories with limited training data [26]. Recent work explores two primary approaches: meta-learning and transfer learning. Meta-learning-based methods focus on acquiring generalizable features from a set of `base` classes, which can then be applied to identify `novel` classes. For example, Kang et al. [23] proposes a technique that re-weights features from `base` classes to predict `novel` classes. Xiao et al. [56] propose a framework addressing both few-shot object detection and few-shot viewpoint estimation. Fan et al. [9] introduces a general FSOD network that learns a matching metric between image pairs, while Wu et al. [53] enhances object features using a universal prototype. More recently, Xu et al. [58] propose a generative approach that is robust to noisy object proposals for `novel` classes. In contrast, transfer learning involves partially freezing network weights pretrained on a `base` dataset to improve a model's ability to generalize to `novel` classes with limited data. Transfer learning approaches often follow a two-stage fine-tuning strategy: first train on `base` classes and then fine-tune the box classifier and regressor with $K$-shots from `novel` classes. This strategy has historically outperformed meta-learning approaches [49]. Recent work has primarily focused on improving classification performance. Sun et al. [48] utilizes a contrastive proposal encoding loss to encourage instance-level intra-class compactness and inter-class variance. Similarly, Li et al. [29] applies a class margin loss to balance inter and intra-class margins.

**Vision Language Models** are trained on a large-scale collection of weakly-supervised image-text pairs collected from the web. These models embed images and text into a shared space, enabling open-vocabulary detection. Early works adapt VLMs for object detection by either distilling the model's predictions for specific image regions [13, 14] or directly incorporating detection components into frozen [27] or fine-tuned [40, 39, 7] encoders. In contrast, RegionCLIP [64] employs a multi-stage training approach, which involves generating pseudo-labels from captioning data, conducting region-text contrastive pre-training, and fine-tuning on detection data. GLIP [30] uses a single text query for the entire image and frames detection as a phrase grounding problem. More recently, Detic [67] addresses long-tail detection performance by leveraging image-level supervision. In the context of open-vocabulary detection, there may be some overlap between categories seen during training and testing. We use the term "zero-shot" when a model has never been trained on the target dataset.

**Fine-Tuning Foundation Models** is of significant interest across many application areas [20, 62, 11]. Standard fine-tuning procedures employ both linear probing [4, 17, 18] and full-finetuning [50, 55, 25, 34] to adapt models to downstream tasks. However, such methods may be suboptimal as they can be computationally expensive. Instead, recent works like CLIP-Adapter [11] and Tip-Adapter [63] fine-tune CLIP using parameter-efficient methods [19, 61, 21] which optimize lightweight MLPs while keeping the encoder frozen. Similarly, inspired by the success of prefix-tuning in language models [6, 22, 16, 12], prompt adaptation [35, 69, 57, 65] replaces hand-crafted prompts like "a photo of a {cls}" with learned word embeddings. CoOp [66] and other prompting methods [35, 69, 65] finetune CLIP via prefix-tuning. Different from most prior work, we investigate fine-tuning strategies for VLM-based detectors using few-shot *multi-modal* examples.

# 3 Foundational FSOD Benchmark

As shown in Fig 2, our proposed Foundational FSOD benchmark utilizes vision-language models (VLMs) pre-trained on diverse, large-scale datasets, which are then aligned to $K$ examples of each target class. We contrast our proposed setup with standard benchmarks and present simple strategies for fine-tuning VLMs below.

## 3.1 Foundational FSOD Benchmark

Existing FSOD benchmarks repurpose well-established datasets like PASCAL VOC [8] and COCO [31] by partitioning them into `base` and `novel` classes for pre-training and fine-tuning, respectively. For COCO, the 60 categories disjoint with PASCAL VOC are used as `base` classes and the remaining 20 are used as `novel` classes [49]. However, this setup is artificial and does not reflect how FSOD is deployed in practice. First, FSOD benchmarks construct a set of `novel` classes that include common concepts such as `car` and `person`, and require FSOD methods to detect these common classes by assuming they have only few examples. Importantly, VLMs like GroundingDINO [33] can already detect common categories with high accuracy on COCO *without fine-tuning* (cf. Table 1). Therefore, we focus on benchmarking Foundational FSOD on more realistic and challenging datasets like nuImages [2]. Second, existing FSOD benchmarks require that datasets are partitioned into `base` and `novel` classes, which is infeasible for large-scale (often private) foundational datasets. For example, although CLIP's [44] model weights are publicly available, its pre-training dataset is not. Instead, FSOD methods should only fine-tune VLMs on $K$-shot annotations for $C$ target classes (or `novel`, as termed in conventional FSOD benchmarks), and also evaluate performance on these $C$ classes.

## 3.2 Few-Shot Multi-Modal Concept Alignment

Although VLMs achieve strong zero-shot performance on common classes, they struggle when the target class is different from concepts encountered on the web (cf. Fig. 1). For example, nuImages [2] defines the cab of a `truck` as a separate concept from its `trailer`. However, Detic [67] detects the entire vehicle as `truck`. This fine-grained distinction is provided to human annotators with visual examples and textual descriptions. We explore seven methods for alignment (either explicitly by updating model weights via gradient-based fine-tuning or in-context via prompting) below.

**Prompt Engineering** uses expressive descriptions, attributes [38], or synonyms [41, 42] in the text prompt to manually improve the alignment of foundation model outputs to target concepts of interest. In our case, we prompt VLMs with synonyms of the nuImages classes to improve detection accuracy. For example, we augment the language query for `pushable-pullable` with synonyms like `cart` and `wheel barrow`. We provide a full list of synonyms in Table 6.

**Standard Fine-Tuning** updates the last few layers of a model to adapt to new target classes. For two-stage object detectors, this typically requires training the box regression and classifier head. However, we find that standard fine-tuning is sub-optimal, motivating our proposed approach below.

**Federated Fine-Tuning** leverages a simple but evidently underappreciated observation: few-shot object detection datasets are actually federated datasets [15]. A federated dataset is comprised of smaller mini-datasets, where each mini-dataset is exhaustively annotated for only a single category. For example, `cars` may or may not appear in the background of the $K$ images annotated with `motorcycles`. However, existing FSOD methods incorrectly assume that no `cars` are present in the background of non-`car` images. We devise a simple loss that incorporates this insight, discussed further in the supplement.

**Language Prompt Tuning** is an established parameter-efficient strategy [47, 28] for updating text embeddings with few-shot examples via fine-tuning. Concretely, for a given language query (e.g. `stroller`), we first extract a text embedding $P^0$ and only fine-tune the text embedding [30].

**Visual Prompting** uses images of target concepts that are difficult to describe through text as prompts to learn novel concepts in-context. For example, although `debris` is a difficult catchall category to define through text, we can use image examples to improve concept alignment. Typically, visual prompts are tokenized and fed as inputs to a frozen VLM.

**Multi-Modal Prompting** combines language and visual prompting to leverage multi-modal features. Intuitively, multi-modal cues can yield better alignment than uni-modal cues alone; in the above

Table 1: **VLM Zero-Shot Inference Is a Strong FSOD Baseline.** Zero-shot inference with VLMs like GroundingDINO resoundingly outperforms state-of-the-art FSOD methods on the COCO FSOD benchmark, motivating the need to re-frame FSOD to embrace foundation models.

| Approach | 30-shots | | |
| --- | --- | --- | --- |
| | AP | Base AP | Novel AP |
| FRCN-ft-full [60] | 18.6 | 20.6 | 12.5 |
| FRCN-BCE [60] | 30.2 | 36.8 | 10.3 |
| TFA w/ fc [49] | 29.3 | 34.5 | 13.5 |
| TFA w/cos [49] | 29.9 | 35.3 | 13.6 |
| MPSR [54] | 17.1 | 18.1 | 14.1 |
| Meta-RCNN [60] | 7.8 | 7.1 | 9.1 |
| FsDetView [56] | 10.0 | 9.3 | 12.0 |
| Retentive R-CNN [10] | 32.9 | 39.3 | 13.8 |
| DiGeo [36] | 33.1 | 39.4 | 14.2 |
| **GroundingDINO (Zero-Shot)** [33] | **48.3** | **46.3** | **54.3** |

case, ambiguous concepts such as `debris` can be clarified with both textual descriptions (e.g `trash can` and `tree branch`) and visual examples (similar to the multi-modal annotator instructions in Fig. 1!). Here, visual and language prompts are tokenized and separately fed as inputs to a frozen VLM. Specifically, MQDet [59] introduces a lightweight Gated Class Scalable Perceiver module that fuses visual cues and text descriptions in the text encoder via class-wise cross attention layers.

**Multi-Modal Chat Assistants** can accomplish many of the same tasks as multi-modal prompting through a multi-modal turn-by-turn conversational interface. However, unlike multi-modal prompting, conversational interfaces allow users to iteratively refine concept definitions, similar to how human annotators often require several rounds of feedback to fully understand the target concept.

## 4 Experiments

We conduct extensive experiments to validate that zero-shot inference from VLMs significantly improves over state-of-the-art FSOD approaches, suggesting that existing benchmarks should be re-framed to allow foundation models to "enter the conversation". Moreover, we demonstrate that leveraging language cues, especially those available for free (e.g., class names), are crucial to improving performance on data-constrained tasks like FSOD.

**Datasets and Metrics.** We repurpose nuImages [2] to support the study of Foundational FSOD. This dataset annotates 18 classes, which are divided into groups with `many`, `medium`, and `few` examples [43, 37]. We report average precision (AP) for each cohort. Although this dataset is not traditionally used for FSOD, nuImages' open-world categories like `debris` and `pushable-pullable` make it particularly challenging (even for VLMs), and is a realistic benchmark for Foundational FSOD. We follow the $K$-shot dataset creation process established by [49], described below. To construct a $K$-shot dataset, we select a target class $c$ and an image at random. If the total annotations for class $c$ in the image are less than or equal to $K$, we add the image to our dataset. We repeat this process for all classes until we have exactly $K$ annotations per class. Since the specific $K$ examples can have a significant impact on the overall performance, we run each experiment over three random data splits and report the average.

### 4.1 Zero-Shot Inference Is A Strong FSOD Baseline

We compare state-of-the-art FSOD methods with zero-shot inference from GroundingDINO [33] on COCO in Table 1. Surprisingly, GroundingDINO outperforms DiGeo [36] by 16.2% AP averaged across both `base` and `novel` categories despite never being trained on COCO images. GroundingDINO's impressive performance is due to its large-scale multi-modal pre-training on Objects365 [45], GoldG [24] and Cap4M [30]. It is worth noting that GroundingDINO achieves higher AP on `novel` classes than `base`, suggesting that `novel` classes in existing benchmarks (e.g., `car` and `person`) are actually not rare in the real world. Therefore, FSOD benchmarks should be re-framed to reflect real-world applications, motivating our setup.

Table 2: **Impact of Large-Scale Pre-Training and Language.** We repurpose nuImages for FSOD following the dataset creation process established by [49]. We group categories by frequency into `many`, `medium` and `few` examples per class [43, 37]. We fine-tune TFA on $K$ examples, but find low performance, $< 3$AP. However, by simply pre-training on more data (LVIS, COCO and ImageNet-21K) and leveraging language cues via a CLIP classifier, 5-shot performance improves from 2.02 AP to 15.12 AP. However, rare (or `few`) classes like `strollers`, `pushable-pullable`, and `debris` remain challenging, motivating our task of VLM alignment.

| Approach | Average Precision (AP) | | | |
| --- | --- | --- | --- | --- |
| | All | Many | Medium | Few |
| **5-shot** | | | | |
| TFA [49] w/ `COCO-base` | 1.33 | 2.78 | 1.43 | 0.23 |
| TFA [49] w/ `LVIS-base` | 2.02 | 1.69 | 4.08 | 0.58 |
| TFA [49] w/ `LVIS,IN-21K,` `COCO` + CLIP Classifier | **15.12** | **22.74** | **18.99** | **4.25** |
| **10-shot** | | | | |
| TFA [49] w/ `COCO-base` | 1.21 | 2.55 | 1.19 | 0.31 |
| TFA [49] w/ `LVIS-base` | 2.27 | 2.05 | 4.51 | 0.58 |
| TFA [49] w/ `LVIS,IN-21K,` `COCO` + CLIP Classifier | **16.09** | **25.46** | **20.00** | **3.73** |
| **30-shot** | | | | |
| TFA [49] w/ `COCO-base` | 1.14 | 2.81 | 0.84 | 0.23 |
| TFA [49] w/ `LVIS-base` | 2.23 | 1.48 | 4.98 | 0.45 |
| TFA [49] w/ `LVIS,IN-21K,` `COCO` + CLIP Classifier | **17.22** | **25.98** | **21.64** | **4.78** |

## 4.2 Foundational FSOD with nuImages

In the context of foundational models, we argue that partitioning datasets into `base` and `novel` classes no longer makes sense. Instead, FSOD methods should only fine-tune on $K$-shot annotations for $C$ target classes, and also evaluate performance on these $C$ classes. We pre-train TFA [49] on diverse datasets and fine-tune on $K$ examples per class and highlight model performance in Table 2. We train two variants of TFA trained on `COCO-base` and `LVIS-base` and fine-tune both models on $K$ examples of the nuImages classes. Surprisingly, both variants of TFA achieve less than 3 AP (cf. Table 2). We posit that this is largely due to poor classification performance. Since both LVIS and COCO classes do not significantly overlap with nuImages classes, learning a classifier from few examples is extremely difficult. However, we find that simply re-training TFA with a frozen CLIP-based classifier (similar to Detic) dramatically increases performance, reiterating the utility of language and web-scale pre-training in data-constrained settings.

## 4.3 Empirical Analysis of Results

We evaluate several popular VLMs on the nuImages Foundational FSOD (10-shot) benchmark and present salient insights from Table 3 below.

**Zero-Shot Detection.** Somewhat unsurprisingly, we find that (1) greater pre-training data scale and diversity, along with (2) larger backbones result in better zero-shot performance. Notably, GLIP achieves 17.01% zero-shot performance, surpassing all other methods trained with less data and smaller backbones.

**Prompt Engineering.** We attempt to improve zero-shot performance using synonyms for class names derived from the annotator textual instructions. We see minor improvements (e.g., Detic improves from 14.40 mAP $\rightarrow$ 14.92 mAP), indicating that leveraging rich textual descriptions beyond class names can improve concept alignment.

**Federated Fine-Tuning.** Standard fine-tuning is sub-optimal for FSOD, as all unannotated classes are treated as negatives. Therefore we use our zero-shot predictions to generate pseudo-labels on training images. We extract pseudo-negatives based on these pseudo-labels by identifying classes *not* in each image (by using detector confidence scores), and leverage pseudo-negatives in our fine-tuning. Notably, we improve over Detic's standard fine-tuning by 1.15 mAP (16.09 mAP $\rightarrow$ 17.24 mAP).

Table 3: **Empirical Analysis of Baselines (10-Shot) on our Benchmark**. We evaluate popular VLMs on the nuImages FSOD Benchmark and find that MQ-GLIP performs the best among all baseline models. Notably, it achieves 17.0 mAP zero-shot language-only performance, and achieves 21.4 mAP via zero-shot multi-modal prompting averaged over all classes. We can iteratively prompt GPT-4o for synonyms to describe each of the few-shot examples to expand MQ GLIP's text prompts, further improving performance by 0.6%. Remarkably, our 2024 competition winners further improved performance to 45.4 mAP, beating our best baseline by 23.3%.

| Approach | Backbone | Pre-Train Data | Average Precision (AP) | | | |
|---|---|---|---|---|---|---|
| | | | All | Many | Med | Few |
| **Zero-Shot Detection** | | | | | | |
| RegionCLIP [64] | RN50 | CC3M | 2.50 | 3.20 | 3.80 | 0.40 |
| Detic [67] | SWIN-B | LVIS, COCO, IN-21K | 14.40 | 25.83 | 16.59 | 2.32 |
| GroundingDINO [33] | SWIN-T | Objects365, GoldG, Cap4M | 12.05 | 17.29 | 15.45 | 3.72 |
| GLIP [30] | SWIN-L | FourODs,GoldG,Cap24M | 17.01 | 23.36 | 19.86 | 8.40 |
| MQ-GLIP-Text [59] | SWIN-L | Objects365, FourODs, GoldG, Cap24M | 17.01 | 23.36 | 19.85 | 8.41 |
| **Prompt Engineering** | | | | | | |
| Detic [67] | SWIN-B | LVIS, COCO, IN-21K | 14.92 | 26.48 | 17.29 | 2.53 |
| GLIP [30] | SWIN-L | FourODs, GoldG, Cap24M | 17.15 | 23.82 | 19.36 | 9.02 |
| **Standard Fine-Tuning** | | | | | | |
| RegionCLIP [64] | RN50 | CC3M | 3.86 | 6.08 | 5.13 | 0.54 |
| Detic [67] | SWIN-B | LVIS, COCO, IN-21K | 16.09 | 25.46 | 20 | 3.73 |
| **Federated Fine-Tuning (Ours)** | | | | | | |
| Detic [67] | SWIN-B | LVIS, COCO, IN-21K | 17.24 | 28.07 | 20.71 | 4.18 |
| Detic [67] w/ Prompt Engineering | SWIN-B | LVIS, COCO, IN-21K | 17.71 | 28.46 | 21.14 | 4.75 |
| **Language Prompt Tuning** | | | | | | |
| GLIP [30] | SWIN-L | FourODs,GoldG,Cap24M | 19.41 | 22.18 | **25.16** | **10.39** |
| **Visual Prompting** | | | | | | |
| MQ-GLIP-Image [59] | SWIN-L | Objects365,FourODs,GoldG,Cap24M | 14.07 | 24.39 | 15.89 | 3.34 |
| **Multi-Modal Prompting** | | | | | | |
| MQ-GLIP [59] | SWIN-L | Objects365,FourODs,GoldG,Cap24M | 21.42 | 32.19 | 23.29 | 10.26 |
| **Multi-Modal Chat Assistants** | | | | | | |
| GPT-4o Zero-Shot Classification [1] | *Private* | *Private* | 9.95 | 16.81 | 12.11 | 1.71 |
| MQ-GLIP Iterative Prompting | *Private* | *Private* | **22.03** | **33.42** | 24.72 | 9.41 |
| **CVPR 2024 Competition Results** | | | | | | |
| PHP_hhh | *Private* | *Private* | **45.35** | **64.25** | **53.43** | **20.19** |
| NJUST KMG | SWIN-L | Objects365V2, OpenImageV6, GoldG, V3Det, COCO2014, COCO2017, LVISV1, GRIT, RefCOCO, RefCOCO+, RefCOCOg, gRef-COCO | 32.56 | 50.21 | 34.87 | 15.16 |
| zjyd_cxy_vision | SWIN-L | Objects365V2, COCO2017, LVIS, GoldG, VG, OpenImagesV6, V3Det, PhraseCut, RefCOCO, RefCOCO+, RefCOCOg, gRef-COCO | 31.57 | 46.59 | 33.32 | 17.03 |

**Multi-Modal Prompting.** We observe that Multi-Modal Prompting (`MQ-GLIP`) achieves the best performance (21.42 mAP) out of all open-source methods in Table 3. We attribute this to its large pre-trained dataset, bigger backbone (SWIN-L) and multi-modal prompts used during inference. Notably, the benefit of multi-modal prompts can be seen by comparing `MQ-GLIP` (21.42 mAP) against `MQ-GLIP-Image` (14.07 mAP), which uses visual prompting and `MQ-GLIP-Text` (17.01 mAP), which uses language prompting. Interestingly, `MQ-GLIP` does not require gradient-based fine-tuning, which differs from all existing conventional few-shot methods. Therefore, we posit that future few-shot methods should further explore in-context learning. Just as multi-modal examples aid human annotator alignment, multi-modal prompting significantly improves VLM concept alignment.

**Multi-Modal Chat Agents**. As shown in Figure 3, we explore the idea of iteratively prompting multi-modal chat assistants like ChatGPT to mimic the real-world workflow of human annotators. Given the strong performance of GPT-4o for general visual understanding, we repurpose it for our task by prompting the model to re-classify image patches from Detic's RPN. Specifically, we ask GPT-4o to predict a class and confidence for each image crop. Surprisingly, we observe reasonable performance (9.95 mAP). However, we find that GPT-4o often incorrectly classifies many image crops with high confidence. Therefore, we prompt GPT-4o to generate its own text descriptions of the few-shot examples according to its "web-scale knowledge". Finally, we use the class names, generated text descriptions, and few-shot visual examples to prompt MQDet to predict new instances of target classes in the test-set. We find that expanding MQDet's in-context prompt with class names, ChatGPT generated text descriptions, and few-shot visual examples improves performance by 0.67% (21.42 mAP → 22.09 mAP) over the MQ-GLIP baseline. Interestingly, although `debris` accuracy does not change when prompted with generated text descriptions, `pushable pullable` (3.6 AP → 15.29 AP) and `barrier` (11.6 AP → 15.31 AP) accuracy improve significantly. We posit that this improvement is due to the reduction in confusion (or over-confident incorrect predictions) between `debris` and `pushable-pullable` (and `barrier`). Surprisingly, a top submission to our CVPR challenge also used ChatGPT to generate meaningful text descriptions to improve *concept alignment*.

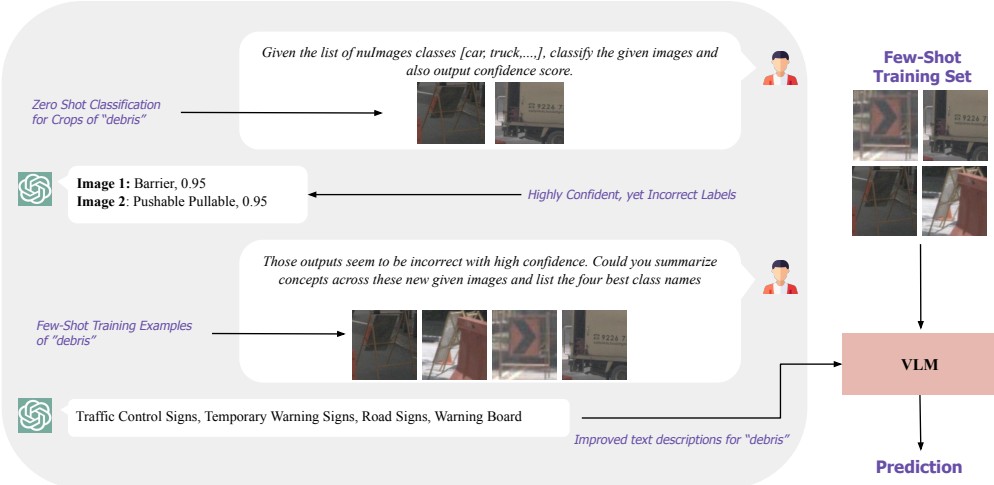

Figure 3: **Iteratively Prompting ChatGPT.** Despite its large-scale pre-training, multi-modal models like ChatGPT-4o also suffers from concept misalignment. Specifically, GPT-4o makes highly confident but incorrect predictions for `debris`. We propose an iterative prompting strategy to better align the model to a target concept. Given a few visual examples per-class from the training-set, we query ChatGPT to use its "web-scale knowledge" to generate text descriptions. We then augment the input to MQDet to incorporate this additional context for zero-shot evaluation.

**CVPR 2024 Challenge**. Our inaugural Foundational FSOD competition (hosted on Eval AI) received submissions from seven teams (some submissions are private). We present a ranked list of participants at the close of our competition on June 7th 2024 AOE in Table 4. Notably, three teams beat our baselines, with the winning team achieving $45.35$ AP! Unfortunately, the top performing team was not willing to publicly share details about their method. We summarize contributions from the other two top teams below.

**NJUST KMG** presents a method that leverages both VLMs and multi-modal chat agents for Foundational FSOD. To address the challenge of misalignment between GroundingDINO and the target concepts of interest, authors generate descriptive referential expressions by prompting ChatGPT to provide descriptive terms for each few-shot instance. The best referential expression for each category is selected by maximizing the Intersection over Union (IoU) between predictions and the ground truth in the few-shot training set. These referential expressions are then used to generate pseudo-labels for all training images. Lastly, GroundingDINO is fine-tuned on a combination of pseudo-labels and ground-truth instances. The full technical report is available here.

**ZJYD CXY Vision** proposes Instruction DINO (ISD), a DETR-based detector architecture that incorporates early fusion of image and text information using a Swin-L visual backbone and EVA02-CLIP-L text encoder. Authors use VLMs like CLIP, TAP, and Llava for negative sample generation (similar to our Federated Fine-Tuning). Authors find that prompt tuning and text encoder fine-tuning generalize better than visual encoder fine-tuning. Similar to NJUST KMG, authors first generate pseudo-label annotations for unlabeled categories before fine-tuning on a combination of pseudo-

Table 4: **CVPR 2024 Foundational FSOD Competition Results**.

| Team Name | Average Precision (AP) | | | |
|---|---|---|---|---|
| | All | Many | Medium | Few |
| PHP_hhh | 45.35 | 64.25 | 53.43 | 20.19 |
| NJUST KMG | 32.56 | 50.21 | 34.87 | 15.16 |
| zjyd_cxy_vision | 31.57 | 46.59 | 33.32 | 17.03 |
| Baseline (MQ-GLIP) | 21.51 | 32.25 | 23.35 | 10.41 |
| team_anon | 17.36 | 25.29 | 21.93 | 5.42 |
| youyouqiu | 13.16 | 11.29 | 19.20 | 7.68 |
| zhao | 11.38 | 11.16 | 16.76 | 5.30 |
| zjdcxy | 7.80 | 5.44 | 13.43 | 3.20 |

Table 5: **Random Split vs "Best" Split**. We construct the "best" split by selecting per-class few-shot examples that lead to the highest performance on a held-out set. Unsurprisingly, the best split performs better than any random split, especially for very limited data settings (e.g., 5-shot detection). This evaluation setting closely mimics how human annotators are "aligned" to target concepts, since annotator guides are constructed using hand-picked iconic visual examples.

| Approach | Average Precision (AP) | | | |
| --- | --- | --- | --- | --- |
| | All | Many | Medium | Few |
| Detic (Zero-Shot) [67] | 14.40 | 25.83 | 16.59 | 2.32 |
| Detic w/ Federated Fine-Tuning *(5-shots, Random Split)* | 16.58 | 27.12 | 19.71 | 4.13 |
| Detic w/ Federated Fine-Tuning *(5-shots, Best Split)* | **18.30** | **28.66** | **21.81** | **5.56** |
| Detic w/ Federated Fine-Tuning *(10-shots, Random Split)* | 17.24 | 28.07 | 20.71 | 4.18 |
| Detic w/ Federated Fine-Tuning *(10-shots, Best Split)* | **18.24** | **28.63** | **22.00** | **5.19** |
| Detic w/ Federated Fine-Tuning *(30-shots, Random Split)* | 18.64 | **29.13** | 22.44 | 5.46 |
| Detic w/ Federated Fine-Tuning *(30-shots, Best Split)* | **18.75** | 28.07 | **23.18** | **5.81** |

labels and ground truth instances. The final method combines prompt tuning and negative sampling, significantly improving mAP. The full technical report is available here.

### 4.4 Analysis of Iconic Few-Shot Images

The specific examples used during few-shot fine-tuning significantly impacts target class performance [49]. However, prior work constructs few-shot splits by randomly sampling $K$ examples per class. In contrast, when creating annotator *instructions*, selecting the right examples to "align" human annotators [3] to subtle aspects of the target concept is carefully considered. To more closely match VLM *concept alignment* with human annotator alignment, we design a simple algorithm to construct the best $K$-shot split for fine-tuning. This allows us to understand which examples are most informative and measure an upper bound in performance.

We construct our *best split* by picking examples corresponding to the best class-wise performance, based on the evaluation of each split on a held-out validation set. For instance, out of 3 random splits for the 5-shot task, one may pick `car` examples from split 1, `bicycle` from split 3 and `debris` from split 2. In Table 5, we observe that the *best-split* performance is always better than its random counterpart. As expected, the choice of examples in 5-shot case is more important than the 30-shot case (1.72 AP difference for 5-shot vs 0.11 AP for 30-shots). We visualize the difference in the splits for `strollers` in nuImages (cf. Figure 4). Unsurprisingly, iconic examples are large and unoccluded.

### 4.5 Limitations and Future Work

Despite using VLMs pre-trained on large-scale datasets, we find that performance for rare categories (defined by the cardinality of each class in the original dataset) is considerably lower than for common

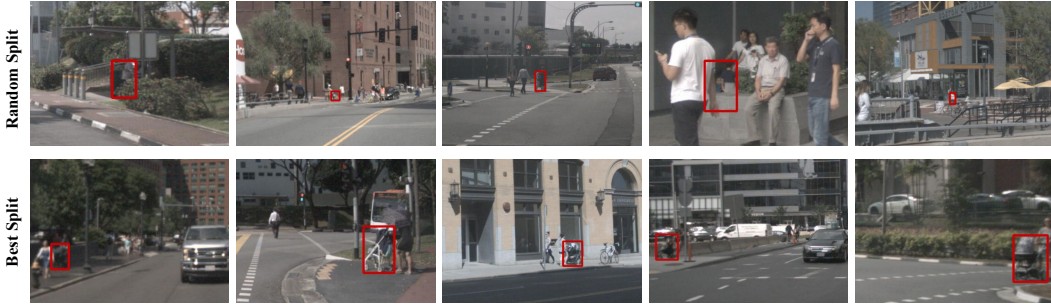

Figure 4: **Visualizing Random and Best Split**. In the top row, we visualize the 5-shot training examples of **strollers** from a *random split*. Similarly, we visualize the 5-shot training examples from the *best split* in the bottom row. We observe that strollers in the *random split* are often occluded, small in size and blurry, making few-shot learning harder. On the other hand, the *best split* examples are larger, have better visual quality and are relatively un-occluded. This visual difference directly translates into better few-shot performance. We achieve **13.09 Stroller AP** for the *random split* and **18.54 Stroller AP** for the *best split*. We show a more comprehensive evaluation in Table 5.

classes. We posit that VLMs are pre-trained with imbalanced data which includes many examples of common categories like `truck` but few examples of rare categories like `stroller` [42]. Our work does not explicitly improve detection performance on rare classes. Interestingly, since VLMs like Detic [67], GLIP [30], and GroundingDINO [33] are trained with different data sources, each model has dramatically different zero-shot performance on novel categories like `stroller`. Ensembling predictions from different VLMs may yield better detection accuracy for rare categories. In addition, although our work motivates the use of rich textual descriptions found in instructions for multi-modal alignment, our current results use only nouns (class names and synonyms) as text prompts.

**Benchmarking in the Era of Foundation Models**. Although we argue that pre-training on large-scale data will be the key enabler for generalization to the open-world, understanding how to appropriately benchmark such methods remains challenging. It is readily accepted that in order to accurately evaluate generalization, one should not train on test data. However, it is difficult to guarantee that foundation models have never seen our specific test data. We address this in our challenge by explicitly prohibiting participants from training on nuImages (and nuScenes). However, should we allow participants to train on similar in-domain data (e.g., other autonomous vehicle datasets such as Argoverse [52])? We argue 'yes'! With enough scale, novel test examples may still be similar to the training set.

**Out-of-Domain Benchmarks**. Another way to address benchmarking is to collect test scenarios that are *designed* to be dissimilar from internet images. For example, out-of-domain images may include medical data (though foundational performance is still surprisingly effective [51]). We admittedly did not take this route, since urban imagery is similar to images found online and arguably many applications of interest fall under this category. Moreover, there exist ample opportunity for technical innovation in this setting (as suggested by our CVPR 2024 challenge results!). Alternatively, one can manually collect and sequester images that will never be released on the internet. Since ensuring privacy may itself be challenging, yet another approach is to leverage the continual learning paradigm [32], where new test sets are continually constructed over time.

**Comparing Models**. Fairly comparing foundation models requires careful consideration. Although accuracy is a valuable metric, it is intrinsically tied to the scale of pre-training data and model architecture. Notably, the LLM community already ranks models via a Pareto frontier of accuracy vs. parameter count. We advocate for a similar approach for Foundational FSOD that considers backbone architecture (e.g., ResNet-50 vs. Swin-B) and pre-training datasets (e.g., CC4M, GoldG, LVIS).

## 5   Conclusion

We revisit few-shot object detection (FSOD) with vision-language models (VLMs) and find that zero-shot inference from web-scale VLMs significantly outperforms leading FSOD methods. However, such foundational models do not fully address few shot recognition because of the *concept alignment* problem; particular concepts in target applications may be different than their use on web-scale datasets. Just as human annotators require concept alignment via multi-modal text and visual examples, we argue that VLMs should be aligned with such few-shot data, formalizing the problem of Foundational FSOD.

## 6   Acknowledgements

This research was supported by Bosch Center for AI (BCAI). We thank BCAI for their financial support and resources, which made this work possible. This work was also supported in part by the Institute of Collaborative Innotation and the University of Macau (SRG2023-00044-FST) and the NSF GRFP (Grant No. DGE2140739).

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
