# A  Baseline Implementation Details

We repurpose nuImages (CC BY-NC-SA 4.0) for all few-shot experiments in the main paper. We evaluate detection performance using $1600 \times 900$ images across 18 classes for all models tested. We create three random splits for each of $K = \{5, 10, 30\}$-shots following the data creation process from [49] and report results averaged across these three seeds. Our test-set is a subset of the (densely annotated) nuImages val-set. We construct our test-set to only include validation images which have at least one annotation from the Few or Medium cohorts (cf. Fig 5). We train all baselines with one RTX 3090 GPU. Our baseline code is available on GitHub and dataset splits are available on HuggingFace.

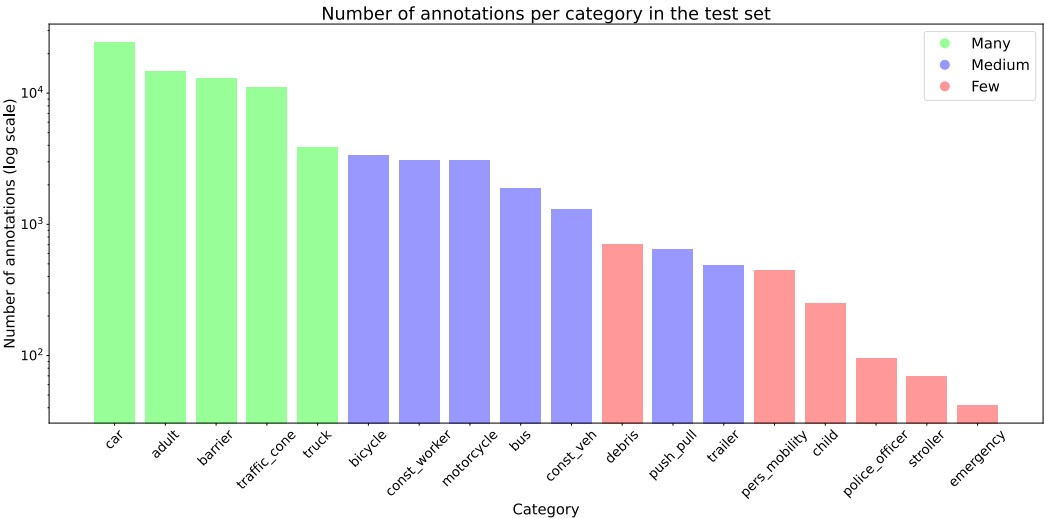

Figure 5: We visualize the distribution of classes in out test-set compared to the cardinalities of classes in the full nuImages val-set. Notably, our sub-sampling strategy of selecting validation images that have at least one annotation from medium or few classes does not significantly alter the true distribution.

**Prompt Engineering**: We leverage rich text descriptions provided by the annotator instructions to select synonyms for each nuImages class. We manually identify the best performing synonyms in Table 6. At test time, we compute the average text embedding of all synonyms to improve classification accuracy.

**Language Prompt Tuning** We train GLIP (SWIN-L backbone) for our prompt tuning experiments for 60 epochs with a learning rate of 0.025, batch size of 4, and weight decay of 0.25.

**Federated Fine-tuning**. We use Detic (Swin-B backbone) pre-trained on LVIS + COCO and ImageNet-21k data for our federated fine-tuning experiments (described in detail in the next section). We use a batch size of 8 and an AdamW optimizer with learning rate of $3.75e - 6$. We fine-tune this model for 8000 iterations on nuImages. We sample 6 categories for each training image, i.e $|S| = 6$ for the FedLoss and InvFedLoss experiments. We derive negatives from pseudolabels with atleast 20% confidence for the Psuedo-Negative experiment.

**Multi-Modal Prompting**. We use MQDet (text-only, vision-only, text + vision for our in-context learning baselines. Unlike the original code base, we tokenize our few shot examples instead of using random queries. Note that zero-shot results for MQ-GLIP-Text and GLIP-L are the same since these models are identical.

# B  Analysis of Federated Fine-Tuning

Prior works follow the $K$-shot dataset creation process established by [49]. Importantly, each image in the dataset is exhaustively annotated for a subset of all classes. Recall, a federated dataset is also comprised of images that are exhaustively annotated for a specific category. This suggests that we can leverage existing insights about federated datasets [15, 68] to train better few-shot object detectors.

Table 6: **Synonyms used for Prompt Engineering.** We manually inspect the nuImages annotator instructions to derive a set of synonyms to improve classification accuracy.

| Original Classes | Class Names with Synonyms |
|---|---|
| `car` | `car` |
| `truck` | `truck`, `pick-up`, `lorry`, `semi-tractor` |
| `construction_vehicle` | `construction_vehicle`, `crane` |
| `bus` | `bus`, `bendy_bus`, `rigid_bus` |
| `trailer` | `trailer` |
| `emergency` | `emergency`, `ambulance`, `police_car`, `police_motorcycle` |
| `motorcycle` | `motorcycle` |
| `bicycle` | `bicycle` |
| `adult` | `adult`, `person` |
| `child` | `child` |
| `police_officer` | `police_officer` |
| `construction_worker` | `construction_worker` |
| `personal_mobility` | `personal_mobility`, `skateboard`, `segway`, `scooter` |
| `stroller` | `stroller` |
| `pushable_pullable` | `pushable_pullable`, `wheel_barrow`, `garbage_bin`, `cart` |
| `barrier` | `barrier`, `K-rail`, `fence`, `bollard`, `guard_rail` |
| `traffic_cone` | `traffic_cone` |
| `debris` | `debris`, `trash_bag` |

**Fine-Tuning with FedLoss**. We fine-tune Detic with Federated Loss (FedLoss) [68] using a subset $S$ of classes $C$ for each training image. Specifically, we use a binary cross-entropy loss on all classes in $S$ and ignore classes outside of $S$ during training. $S$ is comprised of the ground-truth annotation class along with randomly sampled negative classes for each image. We sample these negative classes in proportion to their square-root frequency in the training set. We find that probablistically sampling negatives rather than labeling all unannotated classes as negatives improves fine-tuning results, reliably beating zero-shot performance. Importantly, although FedLoss has been explored in the context of long-tailed detection, applying it to FSOD provides considerable performance improvements, reaffirming that FSOD benchmarks are actually federated datasets.

**Fine-Tuning with Pseudo-Negative Federated Loss (Ours).** Despite the effectiveness of FedLoss, probablistically sampling negatives using dataset-wide statistics is sub-optimal because it does not consider the content of each image. We can improve the accuracy of sampled negatives with pseudo-labels to determine which classes are likely *not* in a particular image. If the maximal score for any class prediction is less than a threshold, we consider this class to be a negative. Using zero-shot model predictions to identify pseudo-negatives yields better results than simply using dataset-wide statistics. We find that this strategy works the best. We present pseudo-code in Alg. 1. All federated fine-tuning results in the main paper are trained with psuedo-negative federated loss.

**Oracle Performance Analysis**. We empirically validate the effectiveness of our pseudo-negative federated loss by computing the upper bound performance when given access to ground-truth negatives and exhaustive annotations for the few-shot data split. Recall, nuImages is exhaustively annotated, but is repurposed for Foundational FSOD.

To compute the set of ground-truth negatives for each image, we use exhaustive ground-truth annotations to determine which categories are not present. Training with ground-truth negatives provides an upper bound on our pseudo-negatives experiment. Next, we train using exhaustive ground-truth annotations to provide an upper bound for the specific set of images used during training. In addition, this experiment highlights the performance gap between having exhaustive negatives and exhaustive annotations.

Table 7 shows that using pseudo-negatives nearly matches the true negative upper bound (16.67 AP vs 16.99 AP). This demonstrates that we are able to reliably estimate negatives in an image, alleviating the problem of learning with sparse annotations. Training with exhaustive annotations

**Algorithm 1:** Psuedo-Negative Federated Loss

```
# Inputs
# img: Randomly Sampled Image
# all_classes: All Classes in Dataset
# gt: Ground Truth Annotations for img
# gt_classes: List of Classes in gt
#
# Outputs
# loss: Psuedo-Negative Federated Loss
#
# Functions
# filter: Returns All Predictions w/
#         Confidence > Threshold
# get_neg: Returns List of Classes Not
#          In Pseudo-Positives
# or: Set Union Operation
# BCE: Binary Cross Entropy Loss

#Step 1: Compute Predictions
#        and Filter by Confidence
pred = Detector(img) # predictions
pseudo_pos = filter(pred, thresh=0.2)

#Step 2: Get Pseudo-Negatives for Image
neg_classes = get_neg(pseudo_pos, all_classes)
select_classes = or(neg_classes, gt_classes)

#Step 3: Compute Deterministic Federated Loss
#        w/ Pseudo-Negatives
loss = 0
for cls in select_classes:
    pred_cls = pred[cls] #predictions for cls
    gt_cls = gt[cls] #ground-truth for cls
    loss += BCE(pred_cls, gt_cls)
return loss
```

yields significantly better results for `many` and `medium` classes. This is unsurprising because the 10-shot FSOD benchmark includes 10 car annotations, while the exhaustively annotated set includes over 550 car annotations!

Despite strong performance on classes with `many` and `medium`, the upper bound for classes with `few` examples remains low (4.21 AP and 3.93 AP). Given the success of training with pseudo-negatives, a natural next-step is to train with pseudo-positives. Our preliminary results suggest that incorporating pseudo-positives does not provide significant improvement over simply training with pseudo-negatives. We posit that training with incorrect pseudo-positives may incur a higher penalty than training with incorrect pseudo-negatives. This is a promising direction for future work.

## C  Impact of Box-Level Supervision for Foundational FSOD

We evaluate the importance of using bounding-box supervised data in pre-training. Unlike Detic, which trains on box-supervised data from `LVIS`, `COCO` and image-text data from `ImageNet21k`, RegionCLIP[64] only pre-trains on image-text pairs from the Conceptual Captions (CC3M) dataset [46]. We report RegionCLIP's zero-shot and fine-tuning performance on nuImages averaged over 3 random splits in Table 8. Detic zero-shot outperforms RegionCLIP zero-shot by $\sim 12$ AP (14.26 vs 2.34). While fine-tuning RegionCLIP improves overall performance, Detic achieves higher accuracy for $K = \{5, 10, 30\}$ shots. This highlights the importance of supervision type (e.g. box-supervised data) and data scale used for pre-training.

Table 7: **Analysis of nuImages Upper Bound Performance**. We compare the accuracy of our proposed approach against upper bounds computed for the FSOD task. Our pseudo-negatives strategy approaches the performance of using ground-truth negatives, demonstrating that pesudo-labels can provide a reliable signal about negatives, especially across classes with `many` and `medium` examples. The performance gap between our best method and exhaustive annotations can be attributed to the large number of additional annotations, particularly for classes with `many` and `medium` examples. Compared to the baseline (14.3 AP), our approach (16.7 AP) closes the gap to the (18.5 AP) upper-bound by over 50%.

| Approach | **10 Shots**: Average Precision (AP) | | | |
| --- | --- | --- | --- | --- |
| | All | Many | Medium | Few |
| Detic (Zero-Shot) [67] | 14.26 | 27.28 | 15.15 | 2.36 |
| + Standard Fine-Tuning | 15.53 | 26.01 | 18.02 | 3.88 |
| w/ FedLoss | 15.57 | 27.20 | 18.13 | 2.89 |
| w/ Pseudo-Negatives | **16.67** | **29.15** | **18.71** | **3.90** |
| w/ True Negatives (*Oracle*) | 16.99 | 29.60 | 18.94 | 4.21 |
| w/ Exhaustive Annotations (*Oracle*) | 18.51 | 33.51 | 20.30 | 3.93 |

Next, we conduct further analysis to diagnose why RegionCLIP zero-shot inference performs so poorly on nuImages (Table 9). RegionCLIP relies on an RPN pre-trained on box-supervised data like `LVIS-base` to extract regions for pre-training. Notably, RegionCLIP (w/ `LVIS-RPN`: 2.34 AP) suffers from poor foreground-vs-background classification compared to Detic. We validate this hypothesis by evaluating RegionCLIP (w/ `GT-RPN`) to measure classification performance. Surprisingly, RegionCLIP achieves significantly higher accuracy (26.44 AP), confirming that RegionCLIP struggles to distinguish between foreground and background in nuImages. This observation highlights the challenge of working with nuImages categories, further motivating our Foundational FSOD benchmark.

Lastly, we evaluate RegionCLIP's performance with `Detic-RPN`. Notably, we observe that the performance improves over RegionCLIP w/ `LVIS-RPN` demonstrating that reducing the number of false positive proposals yields better performance. Furthermore, we filter out low confidence Detic proposals , i.e $< 0.5$ objectness score (w/ `Detic-RPN, 0.5`) and find that this doubles RegionCLIP's zero-shot performance to $7.64$ AP.

## D   NuImages Annotator Instructions

We present an example of the nuImages annotator instructions below. Notably, such annotator instructons are naturally few-shot (e.g. providing a few visual and textual examples describing the target concept), multi-modal, and contain both positive and negative examples. Our proposed Foundational FSOD benchmark, and pseudo-negative federated loss facilitate future work in leveraging rich annotator descriptions, allowing us to "align" VLMs much like how annotators must be "aligned" to subtle aspects of the target class.

Table 8: **RegionCLIP Experiments**. RegionCLIP zero-shot inference performs much worse than Detic. While fine-tuning improves RegionCLIP's performance, it still lags far behind Detic. We posit that this performance difference can be attributed to Detic's box-supervised pre-training and use of language cues from CLIP embeddings.

| Approach | Average Precision (AP) | | | |
| --- | --- | --- | --- | --- |
| | All | Many | Medium | Few |
| RegionCLIP (*Zero-Shot*) [64] | 2.34 | 3.33 | 3.45 | 0.22 |
| Detic (*Zero-Shot*) [67] | **14.26** | **27.28** | **15.15** | **2.36** |
| RegionCLIP (*Fine-Tuning, 5 shots*) [64] | 3.61 | 6.20 | 4.63 | 0.26 |
| Detic (*Fine-Tuning, 5 shots*) [67] | **14.50** | **24.09** | **16.90** | **3.70** |
| RegionCLIP (*Fine-Tuning, 10 shots*) [64] | 3.58 | 6.10 | 4.65 | 0.24 |
| Detic (*Fine-Tuning, 10 shots*) [67] | **15.28** | **26.93** | **18.00** | **3.27** |
| RegionCLIP (*Fine-Tuning, 30 shots*) [64] | 3.57 | 6.13 | 4.61 | 0.22 |
| Detic (*Fine-Tuning, 30 shots*) [67] | **16.65** | **27.45** | **19.46** | **4.02** |

Table 9: **Diagnosing RegionCLIP's Poor Zero-Shot Performance**. RegionCLIP's zero-shot performance lags far behind Detic. Using RegionCLIP's classifier on ground-truth region proposals yields high performance, suggesting that RegionCLIP struggles to accurately distinguish between foreground-vs-background.

| Approach | Average Precision (AP) | | | |
|---|---|---|---|---|
| | All | Many | Medium | Few |
| Detic *(Zero-Shot)* [67] | 14.26 | 27.28 | 15.15 | 2.36 |
| GroundingDINO *(Zero-Shot)* [33] | 11.44 | 17.42 | 14.08 | 3.38 |
| RegionCLIP *(Zero-Shot)* w/ LVIS-RPN [64] | 2.34 | 3.33 | 3.45 | 0.22 |
| RegionCLIP *(Zero-Shot)* w/ Detic-RPN [64] | 3.79 | 6.68 | 4.01 | 1.12 |
| RegionCLIP *(Zero-Shot)* w/ Detic-RPN, 0.5 [64] | 7.64 | 12.81 | 8.88 | 1.88 |
| RegionCLIP *(Zero-Shot)* w/ GT-RPN [64] | 26.44 | 45.33 | 32.25 | 3.92 |

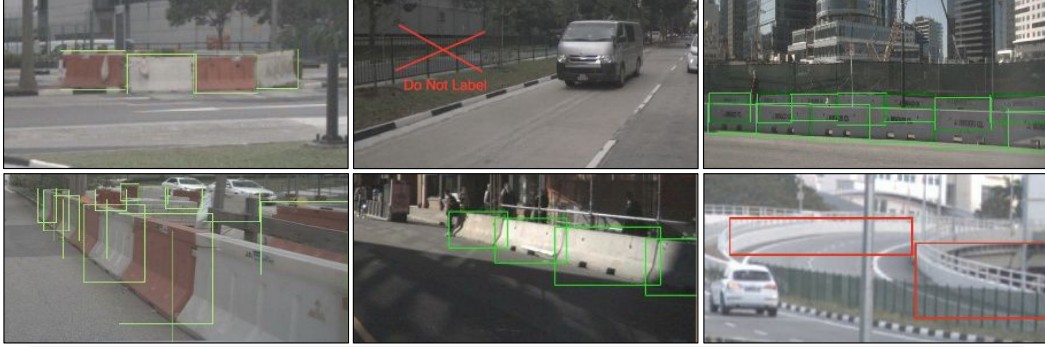

**Barrier**
➔ *Any metal, concrete or water barrier temporarily placed in the scene in order to re-direct vehicle or pedestrian traffic. In particular, includes barriers used at construction zones.*
➔ *If there are multiple barriers either connected or just placed next to each other, they should be annotated separately.*
➔ *If barriers are installed permanently, then do **NOT** include them.*

Figure 6: **NuImages Annotator Instructions.** We include the **multi-modal** annotator instructions `barrier`. Our proposed setup allows FSOD methods to learn such multi-modal examples, similar to how human annotators are taught the labeling policy. Importantly, annotators can also be provided with negative examples (in red) for classes, i.e what **NOT** to label for a certain class. Crucially, our proposed fine-tuning with pseudo-negatives can easily accommodate such negative examples within the proposed setup.

## E   Empirical Analysis of Baselines (5-shot and 30-shot)

We evaluate all baselines for the nuImages experiments with 5-shot and 30-shot in Table 10 and 11, respectively. We find that trends from the main paper hold. Notably, MQ-GLIP with-multi-modal prompting performs the best. However, we find that adding more examples (e.g. MQ-GLIP 5-shot vs. MQ-GLIP 30-shot) does not seem to help in-context learning based methods nearly as much as gradient-based fine-tuning approaches.

## F   Foundational FSOD with LVIS

Although we use nuImages for Foundational FSOD for benchmarking in the main paper and in our competition, other datasets can still be evaluated under this framework. We include benchmarking results for LVIS below. LVIS [15] re-annotates COCO images using 1,230 fine-grained classes, which are divided into frequent, common and rare based on the cardinality of each class. Frequent and common classes are combined to form `LVIS-base` and is used for pre-training. Rare classes are used for `LVIS-novel`. Following [49, 36], we benchmark with LVIS v0.5 on publicly released data splits and report performance averaged across 3 splits for frequent, common, and rare groups ($AP_f$, $AP_c$, $AP_r$) on the LVIS val-set.

As shown in Table 12, Detic outperforms all recent FSOD baselines including DiGeo [36] by ∼6 $AP_c$ & $AP_f$ and achieves 16.3 $AP_r$ without ever seeing any rare class data (e.g., by prompting Detic (`Base Only`) with the rare class names). Importantly, these performance improvements can be attributed to Detic's CLIP-based classifier, which uses CLIP text embeddings corresponding to class

Table 10: **Empirical Analysis of Baselines (5-shot) on nuImages**.

| Approach | Backbone | Pre-Train Data | Average Precision (AP) | | | |
|---|---|---|---|---|---|---|
| | | | All | Many | Med | Few |
| **Zero-Shot Detection** | | | | | | |
| RegionCLIP [64] | RN50 | CC3M | 2.50 | 3.20 | 3.80 | 0.40 |
| Detic [67] | SWIN-B | LVIS, COCO, IN-21K | 14.40 | 25.83 | 16.59 | 2.32 |
| GroundingDINO [33] | SWIN-T | Objects365,GoldG,Cap4M | 12.05 | 17.29 | 15.45 | 3.72 |
| GLIP [30] | SWIN-L | FourODs,GoldG,Cap24M | 17.01 | 23.36 | 19.86 | 8.40 |
| MQ-GLIP-Text [59] | SWIN-L | Objects365,FourODs,GoldG,Cap24M | 17.01 | 23.36 | 19.85 | 8.41 |
| **Prompt Engineering** | | | | | | |
| Detic [67] | SWIN-B | LVIS, COCO, IN-21K | 14.92 | 26.48 | 17.29 | 2.53 |
| GLIP [30] | SWIN-L | FourODs,GoldG,Cap24M | 17.15 | 23.82 | 19.36 | 9.02 |
| **Standard Fine-Tuning** | | | | | | |
| RegionCLIP [64] | RN50 | CC3M | 3.84 | 6.13 | 5.07 | 0.49 |
| Detic [67] | SWIN-B | LVIS, COCO, IN-21K | 15.12 | 22.74 | 18.99 | 4.25 |
| **Federated Fine-Tuning (Ours)** | | | | | | |
| Detic [67] | SWIN-B | LVIS, COCO, IN-21K | 16.58 | 27.12 | 19.71 | 4.13 |
| Detic [67] w/ Prompt Engineering | SWIN-B | LVIS, COCO, IN-21K | 16.96 | 27.89 | 19.94 | 4.37 |
| **Language Prompt Tuning** | | | | | | |
| GLIP [30] | SWIN-L | FourODs,GoldG,Cap24M | 17.79 | 21.07 | 22.87 | 9.12 |
| **Visual Prompting** | | | | | | |
| MQ-GLIP-Image [59] | SWIN-L | Objects365,FourODs,GoldG,Cap24M | 13.42 | 23.05 | 15.00 | 3.54 |
| **Multi-Modal Prompting** | | | | | | |
| MQ-GLIP [59] | SWIN-L | Objects365,FourODs,GoldG,Cap24M | **21.45** | **32.23** | **23.31** | **10.30** |
| **Multi-Modal Chat Assistants** | | | | | | |
| GPT-4o Zero-Shot Classification [1] | *Private* | *Private* | 9.95 | 16.81 | 12.11 | 1.71 |

Table 11: **Empirical Analysis of Baselines (30-shot) on nuImages**.

| Approach | Backbone | Pre-Train Data | Average Precision (AP) | | | |
|---|---|---|---|---|---|---|
| | | | All | Many | Med | Few |
| **Zero-Shot Detection** | | | | | | |
| RegionCLIP [64] | RN50 | CC3M | 2.50 | 3.20 | 3.80 | 0.40 |
| Detic [67] | SWIN-B | LVIS, COCO, IN-21K | 14.40 | 25.83 | 16.59 | 2.32 |
| GroundingDINO [33] | SWIN-T | Objects365,GoldG,Cap4M | 12.05 | 17.29 | 15.45 | 3.72 |
| GLIP [30] | SWIN-L | FourODs,GoldG,Cap24M | 17.01 | 23.36 | 19.86 | 8.40 |
| MQ-GLIP-Text [59] | SWIN-L | Objects365,FourODs,GoldG,Cap24M | 17.01 | 23.36 | 19.85 | 8.41 |
| **Prompt Engineering** | | | | | | |
| Detic [67] | SWIN-B | LVIS, COCO, IN-21K | 14.92 | 26.48 | 17.29 | 2.53 |
| GLIP [30] | SWIN-L | FourODs,GoldG,Cap24M | 17.15 | 23.82 | 19.36 | 9.02 |
| **Standard Fine-Tuning** | | | | | | |
| RegionCLIP [64] | RN50 | CC3M | 3.87 | 6.05 | 5.14 | 0.57 |
| Detic [67] | SWIN-B | LVIS, COCO, IN-21K | 17.22 | 25.98 | 21.64 | 4.78 |
| **Federated Fine-Tuning (Ours)** | | | | | | |
| Detic [67] | SWIN-B | LVIS, COCO, IN-21K | 18.64 | 29.13 | 22.44 | 5.46 |
| Detic [67] w/ Prompt Engineering | SWIN-B | LVIS, COCO, IN-21K | 18.67 | 29.13 | 22.43 | 5.57 |
| **Language Prompt Tuning** | | | | | | |
| GLIP [30] | SWIN-L | FourODs,GoldG,Cap24M | 20.73 | 24.95 | **25.60** | **11.54** |
| **Visual Prompting** | | | | | | |
| MQ-GLIP-Image [59] | SWIN-L | Objects365,FourODs,GoldG,Cap24M | 14.26 | 24.55 | 16.73 | 2.79 |
| **Multi-Modal Prompting** | | | | | | |
| MQ-GLIP [59] | SWIN-L | Objects365,FourODs,GoldG,Cap24M | **21.40** | **32.08** | 23.31 | 10.27 |
| **Multi-Modal Chat Assistants** | | | | | | |
| GPT-4o Zero-Shot Classification [1] | *Private* | *Private* | 9.95 | 16.81 | 12.11 | 1.71 |

names. Such embeddings are a result of large-scale pre-training, which we can effectively leverage for the few-shot task. This highlights the role of language in data-constrained settings.

Further, fine-tuning Detic with pseudo-negatives improves overall performance by 1.6 AP (30.0 vs 31.6) over naive fine-tuning. To contextualize the improvement in performance, we note that between TFA (ICML 2020) and DiGeo (CVPR 2023), the community improved on LVIS FSOD by only 0.5 AP (cf. Table 12). Finally, we note that simply replacing the ResNet-50 backbone with a Swin-B transformer yields a sizeable 12.8 AP improvement for rare classes (19.8 vs. 32.6).

We present fine-tuning results for different variants of Detic on the LVIS 10-shot dataset. Following the standard FSOD protocol, we pre-train Detic on LVIS-base (e.g. frequent and common classes) and fine-tune on 10-shots from each class in LVIS-base and LVIS-novel. Importantly, this means

Table 12: **LVIS Foundational FSOD Performance**. We present fine-tuning results for different variants of Detic on the LVIS 10-shot dataset. We follow the standard FSOD setup and pre-train Detic on `LVIS-base` for fair comparison with prior work. Detic pre-trained only on `LVIS-base` outperforms specialized methods like TFA and DiGeo by $\sim$6 AP, *without fine-tuning* on rare classes. Since we keep the model backbone (ResNet-50) and pre-training data same for all methods, these performance improvements can be attributed to Detic's CLIP-based classifier. This demonstrates that concept leakage through language significantly improve FSOD, and leveraging language cues should be embraced in data constrained settings. Naively fine-tuning Detic yields a performance drop of $AP_f$ and $AP_c$ because treating common classes as negatives in rare category federated datasets hurts performance. Instead, we find that embracing the federated nature of FSOD datasets provides consistent improvements in fine-tuning (30.0 vs. 30.8 for ResNet-50). Further, pseudo-labeling negatives in each image provides a modest improvement (30.8 vs. 31.6 for ResNet-50). Similar trends hold for the SWIN-B and SWIN-L backbones.

| Approach | 10-shots | | | |
| --- | --- | --- | --- | --- |
| | $AP$ | $AP_f$ | $AP_c$ | $AP_r$ |
| **ResNet-50 Backbone** | | | | |
| TFA w/ fc [49] | 24.1 | 27.9 | 23.9 | 14.9 |
| TFA w/ cos [49] | 24.4 | 27.7 | 24.3 | 16.9 |
| DiGeo [36] | 24.9 | 28.5 | 24.6 | 17.3 |
| Detic (Base Only) [67] | 30.0 | 34.4 | 30.8 | 16.3 |
| + Fine-Tuning (Base + Novel) | 30.0 | 33.2 | 31.9 | 15.5 |
| w/ FedLoss | 30.8 | 33.9 | 32.7 | 17.4 |
| w/ Pseudo-Negatives | **31.6** | **34.8** | **32.8** | **19.8** |
| **Swin Backbone** | | | | |
| Detic (Base Only, SWIN-B) [67] | 35.2 | 38.7 | 36.8 | 21.4 |
| + Fine-Tuning (Base + Novel) | 35.9 | 37.1 | 37.8 | 26.7 |
| w/ FedLoss | 36.5 | 36.7 | 38.3 | 30.4 |
| w/ Pseudo-Negatives | 37.2 | 37.7 | 38.2 | 32.6 |
| MQ-GLIP-Text *(SWIN-L)* | 35.8 | 40.2 | 33.1 | 33.0 |
| MQ-GLIP-Image *(SWIN-L)* | 28.8 | 33.0 | 26.6 | 25.1 |
| MQ-GLIP *(SWIN-L)* | **43.4** | **46.4** | **41.8** | **40.1** |

that only results for $AP_r$ are indicative of true few-shot performance. First, we find that naively fine-tuning Detic on `Base + Novel` yields lower performance for $AP_f$ and $AP_r$. Intuitively, this suggests that ignoring the federated nature of FSOD datasets (e.g. by following the standard practice of assuming common classes are negatives for rare class federated datasets) hurts common class performance (cf. Table 12). Importantly, simply training with FedLoss significantly improves over naive fine-tuning, increasing $AP_r$ by 1.9% (15.5 vs. 17.4) and 3.7% (26.7 vs. 30.4) for the ResNet-50 and Swin backbones respectively. Further, leveraging our proposed negative pseudo-labeling strategy provides further improvements over the naive federated loss, increasing $AP_r$ by another 2.4% (17.4 vs. 19.8) and 3.7% (30.4 vs. 32.6) for the ResNet-50 and Swin backbones respectively. Similar to nuImages, we find that multi-modal prompting with MQ-GLIP performs the best of all baselines tested, significantly improving over MQ-GLIP-Text and MQ-GLIP-Image. We attribute MQ-GLIP's strong performance to its bigger backbone and significantly larger pre-training dataset.

**LVIS v0.5 Detic Experiment Details.** We select Detic with a Resnet-50 backbone for fair comparison with prior work. We pre-train Detic on `LVIS-base` for $90k$ iterations with a batch size of 32 using an AdamW optimizer and a learning rate of $2e$-3. All images are resized to $640 \times 640$ and we also enable Repeat Factor Sampling [15]. Following [49], we sample *up to* 10 shots for each class in LVIS (since all classes may not have 10 examples). We use a batch size of 32, learning rate of $2.5e$-5 for $46k$ iterations. We do not use Repeat Factor Sampling for fine-tuning. We sample 50 categories for each training image, i.e., $|S| = 50$ for the FedLoss experiments. We derive negatives from pseudolabels with at least 20% confidence for the Psuedo-Negative experiment.