# OpenReview forum: "Revisiting Few-Shot Object Detection with Vision-Language Models"
_NeurIPS.cc/2024/Datasets_and_Benchmarks_Track — NeurIPS 2024 Track Datasets and Benchmarks Poster_

### Official Review · Reviewer_cYqG · 2024-06-25
**Few-shot Detection in times of pre-trained models**

**Rating:** 7
**Confidence:** 3
**Correctness:** Yes

**Review:**

Overall I like the idea and the experiments are extensive and well documented. I am somewhat unsure about the conclusion of the authors/ the implication of the work on whether this Foundational FSOD is sufficient to investigate the few-shot OD abilities of a model, as it seems to only incorporate some aspects of the previous idea of FSOD.

**Strengths:**

The motivation of investigating FSOD in the context of pretrained models is interesting and relevant.

The experiments appear correct and of large scale (i.e., investigated several different models).

The authors provide extensive discussions of their findings.

Overall, it seems the authors are suggesting to change the previous form of FSOD such that one does not require "unseen" vs "seen" data classes. This is very relevant given the pretrained nature of LMs. However, I am unsure about the conclusion of the authors to only focus on evaluating a model's in-class generalisation rather than its out-of-class generalisation abilities. I wonder if the authors can give some thoughts on how to evaluate this setting with such models?

**Additional Feedback:**

N/A

**Clarity:**

It is somewhat well written. As previously written, I found it hard to understand the difference between the proposed dataset to existing ones and I think the paper could be improved by making this more explicit. I.e. it seems the authors are proposing to only evalaute a model for in-class generalisation abilities.

**Documentation:**

The supplementary material appears to provide links to code and dataset.

**Limitations:**

Yes

**Opportunities For Improvement:**

I do not fully understand what the proposed dataset really looks like, particularly what is different from the baseline setup. Figure 2 attempts to visualise this, but this mainly describes the overall motivation not the proposed dataset. Unless I am missing something section 3.1 seems to be the only relevant section on this. It would be great if you could provide an overview image of the proposed dataset and a more structured textual description of this. But maybe I have just missed some things here so gladly just point me to the relevant parts.

I am a little unsure about the idea of a benchmark here. I.e., in standard settings the point was to compare the performance of different models that are trained on the same data. In this case there is no guarantee that LMs are trained on the same data. How does one handle interpreting the results of Foundational FSOD?

**Relation To Prior Work:**

Yes.

**Summary And Contributions:**

This work investigates the value of existing few shot object detection benchmarks in the context of large pre-trained models that possibly have seen similar data in their pretraining phases. In this context the authors propose a novel protocol to handle the observed issues.

---

> ### Author Rebuttal · Authors · 2024-08-14
>
> Thank you for your review. We address your concerns below.
>
> ### In-Class vs. Out-of-Class Generalization
> We agree that measuring in-class vs out-of-class generalization is difficult in the age of foundation models. Although our target classes of interest (cars, pedestrians, trucks, etc.) have likely been seen before by foundation models trained on web-scale data, we argue that “with enough scale, novel test examples may still be similar to the training set” (L274). Future work should explore benchmarking on out-of-distribution test scenarios like medical imagery that are designed to be dissimilar from internet images. However, foundational performance is still surprisingly effective on such out-of-distribution domains (L278).
>
> ### Details on Proposed Dataset Construction
> We provide a structured textual description of the dataset creation process on L179. Concretely, we construct a K-shot dataset for all nuImages classes following the standard FSOD dataset creation procedure defined by [1].
>
> ### How Can We Interpret the Results of Foundational FSOD when Models Don’t Use the Same Pre-Training Data?
> Great question! Under our setup, models with higher performance are considered better, without regard for their pre-training data. Our benchmarking protocol emphasizes the philosophy of open-world training (L2), i.e., pretrain a foundation model on all possible data and fine tuning it for FSOD. Our approach is similar to that used by the LLM community, where models are also not guaranteed to train using the same data. We highlight several considerations when comparing models in Section 4.5 (“Comparing Models”).
>
> [1] Frustratingly Simple Few-Shot Object Detection. Wang et. al. ICML 2020

---

> > ### Comment · Reviewer_cYqG · 2024-08-19
> >
> > Thank you for the response. I have raised my score.

---

### Official Review · Reviewer_uQBw · 2024-06-25
**Important discussion, but I remain unconvinced by the proposed solution**

**Rating:** 4
**Confidence:** 3
**Clarity:** The paper reads okay, but could use p…

**Review:**

The paper discusses how to evaluate few-shot learners in the age of pre-trained foundation models. See details in the other text fields (I apologize to the authors for this scattering of information, but the review form contains so many fields that it's difficult to be coherent).
* The paper raises many interesting discussion points.
* The proposes a new evaluation pipeline that is more open towards pre-trained models.
* No software/infrastructure appears to be provided.
* I am still not quite convinced by the actual proposed pipeline.

**Strengths:**

The paper highlights an important problem regarding how we are to evaluate the few-shot performance of pre-trained models that may very well have been trained on practically all data on the internet. This makes it difficult to ensure a proper "test" / "training" split. The paper further makes the very valid point that e.g. pre-training a vision backbone on ImageNet (as is very common) also results in a leakage of sorts.

I find it very valuable to have these discussions openly, and I appreciate the efforts by the authors to embrace foundation models in our evaluation pipelines (the alternative seems to be to end up with evaluation pipelines that are not useful for state-of-the-art models).

**Additional Feedback:**

See above.

**Correctness:**

I did not see any immediately incorrect claims in the paper, but I am not entirely convinced that it's okay to forget about the classic test/training split (see above).

**Documentation:**

The paper provides no software or infrastructure that could be documented. I think this is a shame. In the paper's current form, we are all left to reimplement the evaluation protocol from scratch (as far as I can tell) based on the textual description provided in the paper. This strikes me as a brittle process (note: I haven't tried actually doing this, so it may be trivial).

**Ethics:**

I see no ethical concerns

**Limitations:**

The paper raises an important point of discussion: how are we to evaluate foundation scale models, when we often do not know on which data they were trained? The idea of enforcing a clean test/training split quickly becomes so overly stringent that we are unable to evaluate anything. The solution proposed in the paper seems to boil down to the idea of requiring everybody not to train on the nuImages dataset (line 272). I struggle to see how this can be ensured for models trained on non-disclosed proprietary datasets (i.e. most foundation models). What happens when evaluating a model that actually did pre-train on this data?

While I understand the need to revisit the classic test/train splitting, I don't think we can just discard the lessons of cross-validation (which the paper to some extend suggests according to my reading).

**Opportunities For Improvement:**

I think the writing of the paper needs to be somewhat improved: the paper reads fine but it lacks the polish I would expect from a NeurIPS submission. I give examples below:

* The introduction only contains a handful of citation even if it relies on quite a bit of background information. For example, why is GroundingDINO, COCO or nuImages not cited?
* Abbreviations are often not defined, e.g. "mAP", "AP", "TFA". It's fine not to explain these elementary concepts, but at least inform the reader what they are short for.
* LaTeX usage could be improved, e.g. instead of writing "[22] proposes a technique" use the \citet command to get author names; avoid using " and instead use `` and ''; prevent line break of the type "Table<newline>2", etc.

These nitpicking points are all minor, but there's just quite a lot of them ("death by a thousand paper cuts")

As far as I can tell, the paper does not come with code or infrastructure to run the evaluation. I appreciate the textual description of the evaluation pipeline, but such can be brittle. I think it would be valuable to at least have example code on how to perform the evaluation as this is a more precise communication form.

A claimed contribution in the abstract is a discussion of the results of a recently hosted CVPR competition. As far as I could tell, three lines of text are devoted to this in the paper. I honestly think this is too thin: I recommend to either expand the discussion in the paper or take it out altogether.

**Relation To Prior Work:**

I am not a domain expert, but to the best of my knowledge: the paper discusses its relation to prior work.

**Summary And Contributions:**

The paper raises the important discussion of how to evaluate the performance of few-shot learners (here: object detection in images) in the era of large pre-trained models where it is difficult to keep track of what data is used to train models (e.g. often training data is proprietary). The paper describes a potential evaluation protocol.

---

> ### Author Rebuttal · Authors · 2024-08-14
>
> Thank you for your review. We address your concerns below.
>
> ### Links to Software, Dataset, and Documentation
> We do in fact include links to our software and dataset splits in the supplement (Appendix A). Our baseline code has been released on GitHub (https://github.com/anishmadan23/foundational_fsod) and dataset splits are available on HuggingFace (https://huggingface.co/anishmadan23/foundational_fsod/tree/main). Further, our CVPR 2024 challenge evaluation infrastructure is hosted on EvalAI (https://eval.ai/web/challenges/challenge-page/2270/overview). We apologize for not making this more explicit in the main paper, and will update the text accordingly.
>
> ### Discussion on CVPR 2024 Competition
> We discuss the results of our CVPR 2024 competition in Appendix B in the supplementary material. We find that top performers adopt pseudo-labeling strategies (like our federated fine-tuning baseline) to address federated few-shot annotations. Moreover, top performers used VLMs pre-trained on many diverse, large-scale datasets like RefCOCO, Objects 365, LVIS, VG, GoldG, OpenImages, V3Det, and PhraseCOCO. We will move these insights to the main paper.
>
> ### How Did We Ensure Fair Evaluation of Models Trained on Proprietary Datasets In Our CVPR 2024 Challenge (L272)?
> Our solution is to host two submission tracks for methods trained on public datasets vs. undisclosed private datasets. In Table 3 (reproduced below, updated with additional details after the end of the competition), we explicitly highlight the pre-training datasets for each model. Notably, the top performing team does not disclose details about their model or pre-training data. In contrast, the second and third place teams pre-train on publicly available datasets that do not include nuImages or nuScenes. Allowing for two tracks ensures that even foundation models trained on proprietary data can “enter the conversation”. Similar to the LLM community's expectations to not train on agreed-upon test datasets like MMLU, we expect that good faith submissions will not train on the test-set. However, we acknowledge that it is possible that publicly available pre-training datasets will become large enough to make it difficult to guarantee clean train-test splits in the future. We highlight several solutions like manually collecting and sequestering images that will never be released on the internet, or adopting a continually evolving test-set when evaluating foundation models in Section 4.5 (“Out-of-Domain Benchmarks”).
>
> | Team Name       | Backbone | Pre-Train Data                                                                                                    | All   | Many  | Med.  | Few   |
> |-----------------|----------|-------------------------------------------------------------------------------------------------------------------|-------|-------|-------|-------|
> | PHP hhh         | Private  | Private                                                                                                           | 45.35 | 64.25 | 55.43 | 20.19 |
> | NJUST KMG       | SWIN-L   | Objects365V2, OpenImageV6, GoldG, V3Det, COCO2014, COCO2017, LVISV1, GRIT, RefCOCO, RefCOCO+, RefCOCOg, gRef-COCO | 32.56 | 50.21 | 34.87 | 15.16 |
> | ZJYD CXY Vision | SWIN-L   | Objects365V2, COCO2017, LVIS, GoldG, VG, OpenImagesV6, V3Det, PhraseCut, RefCOCO, RefCOCO+, RefCOCOg, gRef-COCO   | 31.57 | 46.59 | 33.32 | 17.03 |
>
> ### Polishing the Paper
> Thanks for the valuable suggestions. We will address these changes in the updated submission.

---

> > ### Comment · Reviewer_uQBw · 2024-08-15
> > **Thanks for the follow-up**
> >
> > Thanks for the informative and useful rebuttal.
> >
> > # Links...
> > Thanks for the relevant links. I had not consulted the supplementary material, so I am happy to hear that this will be moved to the main paper.
> >
> > # Discussion...
> > Again, I am happy to hear that this will be moved to the main paper.
> >
> > # How did we ensure...
> > I appreciate the explanations given here, and I agree that it is reasonable to expect people to abide by explicit statements on "don't train on the test set". I appreciate the idea of collecting non-internet data /continuously evolving test data. Is there any progress or plans on such ideas? I understand that such would be a big undertaking, but it would also hold significant value.
> >
> > # Novelty
> > I have an additional question that only appeared after your rebuttal. What is the novelty of this benchmark paper over your previous competition? Are there changes relative to the competition or should I rather think of the present paper as a mechanism to turn the competition into a citable paper (which would be fair) ?

---

> > > ### Author Response · Authors · 2024-08-15
> > >
> > > Thanks for your response. We address your questions below.
> > >
> > > ### How did we ensure...
> > > Yes, we plan to incorporate a continuously evolving test set in the second iteration of the Foundational FSOD challenge at CVPR 2025. Specifically, we will add aerial images from DOTA-v2 [1] and multi-domain images from RoboFlow 100 [2] to our existing nuImages test-set since these datasets, to the best of our knowledge, are not typically used in foundational pre-training.
> > >
> > > ### Novelty
> > > Our benchmark paper presents a novel protocol, baselines, and summarizes findings from our competition, while the challenge aims to facilitate community adoption and bolster the credibility of our protocol.  Our CVPR 2024 competition is an instantiation of the benchmark protocol presented in this paper. We characterize both efforts as concurrent and complementary contributions.
> > >
> > > [1] Object Detection in Aerial Images: A Large-Scale Benchmark and Challenges. Ding et. al. TPAMI 2021
> > >
> > > [2] RoboFlow 100: A Rich Multi-Domain Object Detection Benchmark. Ciaglia et. al. ArXiv 2022

---

> > > > ### Comment · Reviewer_uQBw · 2024-08-15
> > > > **Thanks - EOM**
> > > >
> > > > Thanks!

---

> > > > > ### Author Response · Authors · 2024-08-23
> > > > >
> > > > > Dear Reviewer uQBw,
> > > > > We'd like to thank you again for your feedback and discussion on our paper. Before the discussion period wraps up, we'd be happy to answer any additional questions you may still have and kindly request that you update your post-rebuttal score if we've addressed your concerns.

---

> > > > > > ### Comment · Reviewer_uQBw · 2024-08-23
> > > > > > **thanks**
> > > > > >
> > > > > > I think I have the information I need at this point. I'll finalize my score during the discussion with the other reviewers.

---

### Official Review · Reviewer_JU26 · 2024-07-25
**Timely and Significant Paper**

**Rating:** 8
**Confidence:** 4
**Correctness:** Yes.
**Clarity:** Yes.

**Review:**

Pros:

1. The paper develops upon the conventional few-shot object detection setting based on the development of VLMs. This is timely and significant to the community.
2. The paper is well-structured and easy to read.
3. The dataset seems well-organized and could be beneficial to future research.
4. The reviewer appreciates the dataset construction in Section 4.4 Analysis of Iconic Few-Shot Images, which has been missing in prior few-shot detection research.

Cons:
1. Although the current dataset is sufficient, more (and also more diverse) downstream domains could provide opportunities for more interesting discoveries.
2. A suggestion is to give some high-level insights for winning solutions for the CVPR 2024 challenge in the main paper.

**Strengths:**

See the review section above.

**Additional Feedback:**

See the review section above.

**Documentation:**

Yes.

**Ethics:**

Yes.

**Limitations:**

Yes.

**Opportunities For Improvement:**

See the review section above.

**Relation To Prior Work:**

Yes.

**Summary And Contributions:**

The paper revisits few-shot object detection within the context of vision-language models trained on large web-scale datasets. It highlights that zero-shot VLMs like GroundingDINO outperform state-of-the-art few-shot detectors on the COCO dataset but may be sub-optimal for specific applications like autonomous vehicle perception. The authors propose aligning foundational models to target concepts using a few multi-modal examples (text and visual), introducing a new benchmark protocol called Foundational FSOD. They evaluate several VLMs using the nuImages dataset and provide empirical analysis. The paper also discusses significant performance improvements in the CVPR 2024 Foundational FSOD competition.

---

> ### Author Rebuttal · Authors · 2024-08-14
>
> Thank you for your review. We address your concerns below.
>
> ### Application to More Downstream Domains
> Great suggestion! We agree that future work should explore diverse domains like medical imagery (L276) in the context of our benchmark protocol. We also benchmark on general object detection with LVIS using the Foundational FSOD protocol in Appendix H.
>
> ### Discussion on CVPR 2024 Competition
> We discuss the results of our CVPR 2024 competition in Appendix B. We find that top performers adopt pseudo-labeling strategies (like our federated fine-tuning baseline) to address federated few-shot annotations. Moreover, top performers used VLMs pre-trained on many diverse, large-scale datasets like RefCOCO, Objects 365, LVIS, VG, GoldG, OpenImages, V3Det, and PhraseCOCO. We will move these insights to the main paper.

---

### Author Rebuttal · Authors · 2024-08-14

We are thrilled that reviewers find that our work is “timely and significant to the community” (JU26), and “raises many interesting discussion points” (uQBw). Moreover, reviewers appreciate that our “experiments are extensive and well documented” (cYqG). We address reviewer concerns below.

- Reviewer uQBw requests links to our software and dataset splits. We do in fact include links to our software and dataset splits in the supplement (Appendix A). Our baseline code has been released on GitHub (https://github.com/anishmadan23/foundational_fsod) and dataset splits are available on HuggingFace (https://huggingface.co/anishmadan23/foundational_fsod/tree/main). Further, our CVPR 2024 challenge evaluation infrastructure is hosted on EvalAI (https://eval.ai/web/challenges/challenge-page/2270/overview). We apologize for not making this more explicit in the main paper, and will update the text accordingly.
- Reviewers JU26 and uQBw request a discussion of our CVPR 2024 competition. We discuss the results of our CVPR 2024 competition in Appendix B. We find that top performers adopt pseudo-labeling strategies (like our federated fine-tuning baseline) to address federated few-shot annotations. Moreover, top performers used VLMs pre-trained on many diverse, large-scale datasets like RefCOCO, Objects 365, LVIS, VG, GoldG, OpenImages, V3Det, and PhraseCOCO. We will move these insights to the main paper.
- Reviewer cYqG requests clarification on evaluating in-class-vs. out-of-class generalization. We agree that measuring in-class vs out-of-class generalization is difficult in the age of foundation models. Although our target classes of interest (cars, pedestrians, trucks, etc.) have likely been seen before by foundation models trained on web-scale data, we argue that “with enough scale, novel test examples may still be similar to the training set” (L274). Future work should explore benchmarking on out-of-distribution test scenarios like medical imagery that are designed to be dissimilar from internet images. However, foundational performance is still surprisingly effective on such out-of-distribution domains (L278).

Updates to the Paper
- JU26, uQBw: We will highlight insights from our CVPR 2024 competition in the main paper.
- uQBw: We will update our paper to make it easier to find links to our software and dataset splits..
- cYqG: We will clarify our evaluation setup in Figure 2
- uQBw: We will incorporate reviewers' suggestions to polish our paper.

---

### Decision · Program_Chairs · 2024-09-26

**Decision:**

Accept (Poster)

**Comment:**

This paper initially received mixed review scores: 8, 4, 6. Reviewers overall recognized the merit of this work, regarding the study of few-shot object detection evaluation timely and significant, the dataset well-organized, and the paper clearly written. They also raised some concerns (mainly from the 2nd Reviewer uQBw): writing lacking polish, code to run evaluation, discussion of the hosted CVPR competition, how to ensure fair evaluation, clarity improvement, etc. The rebuttal was persuasive and addressed most concerns. After rebuttal, the third Reviewer cYqG increased the score to 7. And according to the discussion, uQBw’s concerns are also mostly addressed. But it seems uQBw didn’t get a chance to update the review score. The AC checked the paper, rebuttal, and review comments, and regards the final reviews overall positive. Hence the AC recommends accepting the paper.